# The Influence of Gender Equity in Nursing Education Programs on Nurse Job Satisfaction

**DOI:** 10.3390/healthcare11091318

**Published:** 2023-05-04

**Authors:** Joohee Shim, Da-In Park

**Affiliations:** 1College of Nursing, Yeungnam University College, Daegu 42415, Republic of Korea; jhshim@ync.ac.kr; 2Department of Nursing, College of Life Science and Nano Technology, Hannam University, Daejeon 34430, Republic of Korea

**Keywords:** gender equity, nursing education, male nurses, job satisfaction

## Abstract

(1) Background: One of the strategies to overcome the shortage of nurses is to minimize gender inequity in nursing culture, starting from the undergraduate education program. Although the number of men entering the nursing profession has increased over the years, the portion of male nurses remains low, particularly in Asian countries. Only a few studies have been conducted to identify gender inequity in nursing, and most of these studies used qualitative study design. Therefore, it is necessary to quantitatively identify gender equity in nursing education and the influence it has on nurse job satisfaction. (2) Methods: A total of 165 male nurses participated in this study. Validated questionnaires were used to assess gender equity in nursing education programs, nurse job satisfaction, nurse job esteem, and nursing professional pride. (3) Results: The mean score of gender equity in nursing programs was 62.6, showing a positive correlation with nurse job satisfaction, nurse job esteem, and nursing professional pride. Gender equity positively predicted nurse job esteem. (4) Conclusions: The gender equity in nursing education programs has positive predictive effects on the factors that influence male nurse job satisfaction. In order to increase nurse job retention, educators should incorporate strategies to minimize gender inequity.

## 1. Introduction

The global shortage of nurses has been a challenge for healthcare systems. This took the highest toll with the recent COVID-19 pandemic. Needless to say, the shortage of nurses has a direct negative influence on the quality of patient care and patient outcomes [1]. To resolve these nursing shortage problems, nursing researchers and educators have suggested increasing the proportion of male nurses and minimizing gender imbalance in nursing education programs [2]. Even though the number of men entering the nursing profession has increased in recent decades, the number of male nurses remains low. According to a recent report, only 5 to 10% of the nursing workforce is comprised of men [3]. Additionally, on average, the turnover rate for male nurses is twice as high as that of female nurses, and they typically change jobs within four years of starting their nursing career [4].

The low portion of the male nursing workforce is more prominent in Asian countries compared with that of Western countries [2,5]; this may be due to cultural differences and stronger social discrimination and prejudices towards male nurses [6]. In China, the number of active male nurses is far lower than female nurses; this is largely due to low salaries and the gender bias towards male nurses [7]. A Taiwanese study also found that social discrimination towards male nurses among the public, patients, and co-workers created frustration and intention to leave clinical nursing jobs [7]. Gender inequity and gender stereotypes may be the reason for the low percentage of male nurses, and several studies have reported that gender bias hinders gender diversity in nursing professions [7,8,9]. Since Florence Nightingale, who believed that nursing was a natural extension of virtuous womanhood, nursing has consistently been a female-dominated profession. This has caused difficulties for males who wish to become nurses, and this gender bias makes it difficult for male nursing students to enter nursing colleges, as well as to decide on a career [4]. In addition, male nurses’ caring, empathy, and intuition abilities are often ignored largely due to the prejudice that caring, the essence of nursing, is a female characteristic [10]. Therefore, patients expect female nurses to take care of them, which further undermines the attitudes of male and they experience gender stereotypes. Taken together, the role of men in nursing has been a continuing concern for gender and occupational scholars the examining processes that maintain or challenge professional gender segregation [11]. Therefore, the factors for the low ratio of male nurses should be identified first in order to increase the number of male nurses and thus the shortage of nursing manpower in the future.

Numerous studies have indicated that job satisfaction is particularly important for increasing the retention rates of nurses [12]. Nevertheless, increased job satisfaction for nurses results in increased nurse motivation and a sense of responsibility, which is then more likely to result in nurse retention [13]. In addition, job satisfaction is an important factor for both patients and nurses as it is related to the improvement in nursing quality [14] and positive patient outcome [15]. Therefore, in order to solve the problem of insufficient nursing manpower, further research is needed to identify the factors influencing nursing job satisfaction. Nursing professionalism is defined as the sum of the nurse’s beliefs, notions, and impressions about nursing as a profession [16], and professional pride refers to the degree of ability to reflect a personal evaluation or judgment of a person’s worth that reflects what others think of himself or herself [17]. Nursing professional pride determines the motivation and aspirations in nursing [16] and is closely related to maintaining the quality of patient care [18]. In addition, as professional identity is related to important outcomes such as nurses’ intention to remain and job satisfaction [19], nursing professional pride has a positive effect on the improvement of nurses’ job satisfaction. Lastly, job esteem is the level of respect and authority that an individual believes about their job, as well as beliefs and values associated with the job [20]. As a factor influencing job satisfaction, it is an emotional result from the organization, job, and job environment [21]. Low job esteem can have a negative impact on job attitude, leading to job turnover and career interruption [22], whereas high job esteem has a positive effect on job satisfaction, resulting in high job satisfaction [23]. It is considered that job esteem will act as a factor in nurses’ job satisfaction.

This study was conducted to identify the level of gender equity in nursing programs and the influencing factors of gender equity, pride in the profession of nursing, and job esteem that have an effect on job satisfaction.

## 2. Materials and Methods

### 2.1. Aims

The aims of this study were (a) to understand gender equity in nursing education; (b) to analyze correlation between gender equity in nursing education and job satisfaction, job esteem, and nursing profession pride among male nurses; (c) to determine if nurse job esteem plays an intermediary role between gender equity in nursing education and nurse job satisfaction; and (d) to explore whether the mediating process of gender equity in nursing education affects male nurses’ job satisfaction, job esteem, and nursing professional pride.

### 2.2. Study Design and Participanys

This study was conducted using a descriptive, cross-sectional design. Participants were recruited from seven tertiary hospitals in three different major cities in Korea. Furthermore, potential participants were recruited through a snowballing method. Data were collected from November 2021 to April 2022. The inclusion criteria were as follows: (a) male, (b) at least 18 years of age, (c) graduated and received a bachelor’s degree in nursing from a four-year university or four-year college in Korea, (d) currently holds an active Korean registered nurse (RN) licensure, (e) had at least 6 months of working experience as a clinical RN, and (f) volunteered to participate in this study and signed the informed consent. The exclusion criteria were as follows: (a) female, (b) never worked as a clinical RN or had less than 6 months of working experience as a clinical RN, and (c) graduated and received an associate degree in nursing from a three-year college in Korea. To ensure a sufficient sample size, a priori power analysis of the correlation coefficient was performed using the G*power software version 3.1 with an effect size of 0.15, an alpha value of 0.05, a power of 0.80, and 12 predictors. The calculated minimum sample size was 127. However, considering a 20% possible drop-out rate, the estimated minimum sample size for this study was 153.

### 2.3. Measurement

#### 2.3.1. Sociographic Variables

A simple self-reporting questionnaire was used to collect data regarding the study participants’ sociodemographic information including gender, age, graduated school type (four-year university or four-year college), and length of work experience as a clinical RN.

#### 2.3.2. Gender Equity in Nursing Program

The gender equity in nursing education was assessed using the Inventory of Male Friendliness in Nursing Program (IMFNP) [24]. IMFNP is a 31-item questionnaire that measures the level of male receptiveness and the barriers in nursing education programs [24]. This scale uses a four-point Likert scale, with the total score ranging from 0 to 120. A higher score indicated a more receptive education climate for male students, thus referring to a higher level of gender equity in nursing education. We set the mid-point of 60 as the cut-off point to define low and high levels of gender equity. In addition, we asked the study participants answer questions regarding their experiences with gender equity in nursing prior to their nursing education. These questions were answered based on yes or no, and the included contents were as follows: (a) if their loved ones or themselves received care from a male nurse prior to the start of their nursing education, (b) if they had known any male RN prior to the start of their nursing education, (c) if they had any mentoring program exclusive for male students in their nursing education, (d) if they had any male faculty members teaching their nursing education, and (e) if they had any fellow male students in their graduating class.

#### 2.3.3. Nurse Job Satisfaction

Participants’ level of job satisfaction was assessed using the Job Satisfaction Scale for Clinical Nurses (JSSCN), a 33-item questionnaire [25]. JSSCN was developed specifically for the Korean nurse population and quantifies six domains that affect job satisfaction among clinical nurses. The subdomains include recognition from the organization and professional achievement, personal maturation through the nursing profession, interpersonal interaction with respect and recognition, the accomplishment of accountability as a nurse, display of professional competency, and stability and job worth [25]. Each item is answered on a five-point Likert scale, with a higher total score indicating a higher level of job satisfaction as a nurse. The total possible score ranged from 33 to 165.

#### 2.3.4. Nurse Job Esteem

Job esteem as a nurse was measured using the Job-Esteem Scale for Korean Nurses (JES-KN) [26]. JES-KN consists of six subscales that measure professional self-awareness, professional competence, role, and expertise of care, social trust and respect, respect and recognition of the organization and professional authority, and future value. The scale includes 28 items and uses a five-point Likert scale. The total possible score ranges between 28 and 140 with a higher JES-KEN score referring to a higher level of self-reported job esteem as a nurse.

#### 2.3.5. Nursing Professional Pride

The level of study participants’ pride as a professional nurse was assessed using the Nursing Professional Pride (NPP) [27]. NPP is a 27-item instrument that includes the following six subscales: feeling of vocation, role satisfaction, role of problem solver, self-achievement, and willingness to stay [27]. Each item uses a five-point Likert scale ranging from 1 to 5, and the total score ranges from 27 to 135. A higher NPP score indicates a higher level of pride as a professional nurse.

### 2.4. Data Analysis

General characteristics of the study participants were analyzed using descriptive statistics such as means and SDs for continuous variables, and frequency for categorical variables, as appropriate. The correlations between the study variables were examined using Pearson’s correlation coefficient. Based on Hayes’ PROCESS macro program, the bootstrapping method was used to test the significance of the mediated models. In this study, Model 7 was used among the PROCESS analysis methods, and bootstrapping was designated 5000 times and set to 95% of the confidence interval. During the verification, we intended to prevent the multicollinearity problem between variables by averaging independent and control variables and then creating and injecting interaction terms. All statistical analyses were conducted using SPSS software version 26.0 (IBM Corporation, Armonk, NY, USA). The effects of mediation and moderation were examined through the SPSS macro program Process 3.3. The statistical significance was defined as a two-tailed *p* value of <0.05.

### 2.5. Ethical Considerations

The data collection for this study was conducted after obtaining the ethical approval from the Yeungnam University College’s institutional review board. All questionnaires were anonymously self-reported by the study participants who provided voluntary written informed consent. In addition, participants were provided with a detailed explanation of the study and with the contact information of the researchers. Those who had any inquiries regarding the study or who wished to withdraw from the study could contact the researchers at any time throughout the data collection period.

## 3. Results

### 3.1. Characteristics of Study Participants

The sociodemographic characteristics of the included study participants are shown in Table 1. A total of 165 male nurses with a mean age of 33.0 (SD 4.9) were included in this study. The majority of the study participants received their nursing degree from four-year universities. In addition, most of the participants had 2 to 10 years of working experience as clinical nurses. The average score for the gender equity in nursing education was 62.6 (SD 9.8) with 60% of the participants rating their nursing education to have a high level of gender equity. However, 55.8% of the participants never received care from a male RN, 81.2% never knew a male RN prior to starting their nursing education, 79.4% did not have a mentoring program specifically for male students, and 50.9% did not have any male faculty members. On the other hand, the majority of the participants had other male classmates throughout their nursing education (97%). The participants showed a mean nurse job satisfaction score of 105.5 (SD 12.9), job esteem score of 95.6 (SD 11.9), and professional pride score of 118.1 (SD 15.2) (Table 1). 

### 3.2. The Gender Equity in Nursing Program

Given that the cut-off value for a low level of gender equity in nursing programs is 60, the mean IMFNP score was fairly low with 62.6 (SD 9.8) (Table 1).

The correlation analysis showed that the gender equity in nursing programs was positively associated with nurse job satisfaction (r = 0.239, *p* = 0.002), nurse job esteem (r = 0.330, *p* < 0.001), and nursing professional pride (r = 0.273, *p* < 0.001). In addition, the nurse job satisfaction level showed significant positive correlations with nurse job esteem (r = 0.733, *p* < 0.001) and nursing professional pride (r = 0.837, *p* < 0.001). Lastly, the nurse job esteem also showed statistically significant positive correlational values with nursing professional pride (r = 0.714, *p* < 0.001) (Table 2).

### 3.3. Model Testing

The mediation effect of nurse job esteem on the association between gender equity and nursing professional pride was positive, and as a result, gender equity positively predicted nurse job esteem (β = 0.194, t = 2.64, *p* = 0.009). When the mediation variable of nursing professional pride was added, the positive predictive effect of gender equity on nurse job esteem still showed its significance (β = 0.011, t = 2.43, *p* = 0.016). Furthermore, the upper and lower bounds of the bootstrap 95% confidence intervals of the direct impact of gender equity and nursing professional pride did not contain 0 (Table 3 and Table 4) (Figure 1a).

Furthermore, a simple slope test was conducted to examine the moderating trend of nursing professional pride on the relationship between gender equity in nursing education and nurse job satisfaction by examining the plus and minus standard deviation from the mean of the two variables (Figure 1b). Under the condition of high nursing professional pride, the nurses’ gender equity had a significant predictive effect on nurse job satisfaction, which means the nurses with high professional pride viewed their nursing education program to have high gender equity and were more satisfied with their current nursing job.

## 4. Discussion

This study was conducted to understand gender equity in nursing education and to explore the effects of gender equity on male nurse job satisfaction, nurse job esteem, and nursing professional pride and job esteem. Our results show that the level of gender equity in nursing education and nurse job esteem are moderated by an individual level of nursing professional pride. In addition, in the relationship between gender equity and job satisfaction, the mediating effect of job esteem is affected by the moderating effect of professional pride in nursing.

The level of nurse job esteem showed a significant mediating effect in the relationship between gender equity and job satisfaction. Our results were in accordance with previous studies that showed a positive relationship between nurses’ job esteem and job satisfaction [28,29,30]. Nurses with higher levels of job esteem are more likely to adapt to high stress levels and maintain positive mental health status and establish healthier and more effective stress relief strategies [31]. In addition, nurses with higher job esteem tend to work more effectively, which further increases nurse job satisfaction levels, thereby positively affecting nurse job retention [32]. As a result, it will be possible to increase the effectiveness of the organization by increasing the retention of competent nurses and reducing the cost of staff replacement due to turnover. Undergraduate nursing education may be the most critical time to improve nursing job esteem. Previous studies have shown that the level of nurse job esteem increased among undergraduate students when they achieved satisfactory scores on either didactic or practicum courses [26,30]. Therefore, it is necessary to educate nursing students to improve job esteem, which contributes to higher job satisfaction later on in their career.

Although our results showed gender equity in nursing education had an insignificant effect on nurse job satisfaction directly, this may be due to the instrument used to measure gender equity. IMFNP is a retrospective, self-reporting questionnaire that aims to capture possible gender barriers in the nursing education program and how important those barriers are to an individual. Because of the retrospective nature of the instrument, there may have been underrepresentation of the gender inequity, and thus the study participants may not have shown a significant direct effect on their current job satisfaction.

Stereotypes in nursing have changed only slightly over the years, and there is a socially established gender role concept that nursing is a profession reserved for women [33]. Male nurses report being dissatisfied with their role as nurses for several reasons, including feelings of unappreciation, minority groups, lack of role models, and excessive expectations and stereotypes about physically demanding work [34]. Previous studies conducted in South Africa, Malaysia, and Canada reported that nursing literature, mass media, and gender stereotypes were the main obstacles that prevented males from entering the nursing profession [35,36,37]. The Gender Equity Hub of the Global Health Workforce Network established by the World Health Organization also reported that gender stereotypes often hinder men from entering nursing [38]. These suggest that greater efforts are needed to promote males to enter nursing education programs and to provide education with gender equity. In addition, there needs to be a continuous effort to establish gender balance in the nursing workforce.

As a result of this study, considering the relationship between gender equity and job satisfaction, it was found that the higher the nursing professional pride, the higher the job satisfaction. This is consistent with the fact that Korean and Chinese nurses’ nursing professional pride was found to be a significant factor influencing job satisfaction [39]. Nurses with higher professional pride showed that they were more motivated, had improved professionalism, and had higher job satisfaction [40]. In a previous study comparing male and female nurses, male nurses showed higher ambitions and drive for career development and advancement [41]. In other words, career success and professional pride tend to be considered important, so it can be thought of in connection with the functioning of nursing professional pride as a moderating variable for job satisfaction. In addition, it has been reported that male nurses’ career development and social support are positively related [4]. As our results showed that male nurses tend to have higher job satisfaction when they achieve higher professional pride and job esteem, clinical settings should provide opportunities for these nurses to be more actively engaged in patient care and provide positive reinforcements. We suggest further research is required to improve nursing professional pride.

As life expectancy increases and more people suffer from chronic diseases, the shortage of nursing care becomes a major challenge. In this situation, increasing the job satisfaction of nurses is an important factor for improving patients’ awareness of the quality of medical care and securing appropriate nursing personnel [42]. The task of nursing is to solve the shortage of nursing manpower, and as a solution to this, it is necessary to increase the number of male nurses. To this end, efforts are needed for gender equity in both universities and hospitals.

This study is significant as it provides basic data for increasing male nurses and nursing staff. Through this study, it will be possible to increase the number of male nurses by eliminating gender bias and developing nursing education and culture. Furthermore, it will be able to contribute to the expansion of nursing manpower and securing professional positions in nursing professions.

## 5. Conclusions

This study was conducted to identify the effect on job satisfaction of male nurses and to provide evidence-based data for resolving the shortage of nurses by minimizing gender inequity to increase the number of male nurses. It was found that gender equity improves job satisfaction through job esteem as the nursing professional pride improves. To increase the job retention of male nurses, efforts to achieve gender equity should be started at the undergraduate level. We suggest nursing education programs to actively assess for possible gender barriers, consider methods to avoid gender inequity, and provide guidance and support for future male nurses. In addition, interventional programs that aim to improve male nurses’ professional pride and job esteem should be provided for active male nurses in clinical settings.

## Figures and Tables

**Figure 1 healthcare-11-01318-f001:**
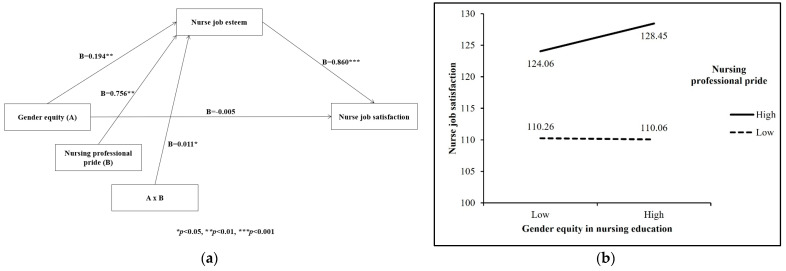
(**a**) Mediated moderating effects between study variables. (**b**) Gender equity in nursing education moderates the relationship between nurse job satisfaction and nursing professional pride.

**Table 1 healthcare-11-01318-t001:** General characteristics of the study participants (N = 165).

Demographic Variables	*n* (%) or M ± SD
Age	33.0 ± 4.9
Graduated school type	
4-year university	147 (89.1)
4-year college	18 (11.9)
Work experience	
6 months–1 year	8 (4.8)
1 year–2 year	22 (13.3)
2 year–5 year	67 (40.6)
5 year–10 year	45 (27.3)
≥10 year	23 (13.9)
Gender equity in nursing program	
IMFNP (possible score range 0–120)	62.6 ± 9.8
Low level (≤60)	66 (40.0)
High level (>60)	99 (60.0)
Received care from a male RN	
Yes	73 (44.2)
No	92 (55.8)
Known male RN	
Yes	31 (18.8)
No	134 (81.2)
Mentoring program for male nursing students	
Yes	34 (20.6)
No	131 (79.4)
Male faculty members	
Yes	81 (49.1)
No	84 (50.9)
Male classmate	
Yes	160 (97.0)
No	5 (3.0)
Nurse job satisfaction (possible score range 33–165)	105.5 ± 12.9
Nurse job esteem (possible score range 28–140)	95.6 ± 11.9
Nursing professional pride (possible score range 28–140)	118.1 ± 15.2

**Table 2 healthcare-11-01318-t002:** Correlation analysis of the study variables (N = 165).

	Nurse Job Satisfaction	Nurse Job Esteem	Nursing Professional Pride
**Gender equity**	0.239 *	0.330 **	0.273 **
**Nurse job satisfaction**	1	0.733 **	0.837 **
**Nurse job esteem**		1	0.714 **
**Nurse professional pride**			1

* *p* < 0.01 (two-tailed) ), ** *p* < 0.001 (two-tailed).

**Table 3 healthcare-11-01318-t003:** Mediated moderating effect of nursing professional pride.

**Variables**	**Dependent Variable Model (Nurse Job Esteem)**					
	**B**	**SE**	**t**	** *p* **	**95% CI**	
					**LLCI ***	**ULCI ****
**Constant**	105.156	0.701	150.00	0.000	103.772	106.541
**Gender equity (A)**	0.194	0.073	2.64	0.009	0.049	0.338
**Nursing professional pride (B)**	0.756	0.061	12.46	0.000	0.636	0.876
**A × B**	0.011	0.005	2.43	0.016	0.002	0.020
**Variables**	**Dependent variable model (nurse job satisfaction)**					
	**B**	**SE**	**t**	* **p** *	**95% CI**	
					**LLCI ***	**ULCI ****
**Constant**	27.315	7.050	3.87	0.000	13.392	41.237
**Gender equity**	−0.005	0.088	−0.06	0.956	−0.179	0.169
**Nurse job esteem**	0.860	0.066	12.96	0.000	0.729	0.991
**Mediated moderation effect**			**B**	**SE**	**95% CI**	
					**LLCI ***	**ULCI ****
**Nursing professional pride**			0.010	0.004	0.002	0.019

* LLCI: B’s lower bound value within the 95% confidence interval, ** ULCI: B’s upper bound value within the 95% confidence interval.

**Table 4 healthcare-11-01318-t004:** Mediated moderating effect of nursing professional pride.

Nursing Professional Pride	Indirect Effects of Mediated Regulation According to the Value of the Nursing Professional Pride			
	B	SE	LLCI *	ULCI **
Mean − 1 SD (−11.964)	0.052	0.087	−0.133	0.214
Mean (0.000)	0.167	0.068	0.032	0.298
Mean + 1 SD (11.964)	0.281	0.080	0.135	0.442

* LLCI: B’s lower bound value within the 95% confidence interval, ** ULCI: B’s upper bound value within the 95% confidence interval.

## Data Availability

The datasets generated for this study are available upon request from the corresponding author.

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
