# Peer review of "The Influence of Gender Equity in Nursing Education Programs on Nurse Job Satisfaction"

_healthcare, 2023, doi:10.3390/healthcare11091318_

Round 1

Reviewer 1 Report

Interesting topic. I believe that this problem is encountered everywhere in the world. The aim of the work is well formulated. Everything is clearly explained in the introduction. The methodology was done in a modern way in accordance with the aim of the work. The presentation of the results is satisfactory. In the discussion, the authors compare their results with relevant studies. The conclusion is interesting and appropriate. There are too many references.

Author Response

Dear Reviewer 1,

Thank you for your valuable comments regarding our manuscript entitled “The influence of gender equity in nursing education program on nurse job satisfaction” to be considered for publication in the special issue “Global Perspectives on Nursing and Midwifery Workforce Development” of Healthcare.

We greatly appreciate your comments. As you have suggested, we omitted some of the references that were redundant after a careful review. The total number of references used for this manuscript has changed from 62 to 46.

We believe that this manuscript is appropriate for publication in the special issue “Global Perspectives on Nursing and Midwifery Workforce Development” of Healthcare, as it provides scientific evidence of gender equity in nursing education program and how it associated with nursing workforce.

The authors of this manuscript comply with the ethics policies of the Healthcare as stated in the Healthcare Author Guidelines on its website. This manuscript has not been published elsewhere nor simultaneously submitted for publication elsewhere. We have no conflicts of interest to disclose. The manuscript has been read and approved for submission by all contributing authors.

Thank you for your time.

Reviewer 2 Report

1.Data was collected between November 2011 and April 2022.Was the data collection period too long, or is this a typo?

2.To make it easier for readers, kindly include the measurement score range for each scale in Table 1. This study uses 60 points as the cut-off point of IMFNP, and divides it into two groups, low and high levels of gender equity. Please present the percentages of the two groups in Table 1.

3.The current manuscript clarifies the association between variables; however, it falls short in providing a clear explanation of the research's implementation and practical recommendations. To improve the paper, it is recommended to include more details on the implementation behind the research and specific suggestions for practical application.

Author Response

Dear Reviewer 2,

Thank you for your valuable comments regarding our manuscript entitled “The influence of gender equity in nursing education program on nurse job satisfaction” to be considered for publication in the special issue “Global Perspectives on Nursing and Midwifery Workforce Development” of Healthcare.

We greatly appreciate your comments. As you have suggested, we have revised our manuscript in the following manner.

  1. We apologize for the typo in the first submitted manuscript. The data collection period was from November 2021 to April 2022. We have revised the method section of the manuscript accordingly.
  2. We appreciate your suggestion regarding Table 1. We have also added the score range for each scale and the number and percentage of low and high levels of gender equity.
  3. Thank you for your comments regarding the discussion and conclusion of the manuscript. We have revised both sections to provide a clear explanation of our research’s implementation and practical recommendations. We provided specific suggestions that could be reinforced in both nursing education programs and clinical settings in the future.

We believe that this manuscript is appropriate for publication in the special issue “Global Perspectives on Nursing and Midwifery Workforce Development” of Healthcare, as it provides scientific evidence of gender equity in nursing education program and how it associated with nursing workforce.

The authors of this manuscript comply with the ethics policies of the Healthcare as stated in the Healthcare Author Guidelines on its website. This manuscript has not been published elsewhere nor simultaneously submitted for publication elsewhere. We have no conflicts of interest to disclose. The manuscript has been read and approved for submission by all contributing authors.

Thank you for your time.

Reviewer 3 Report

The manuscript is about the relation between gender equity , nurse job satisfaction, job esteem and nursing professional pride. The subject is important but there are concerns in methodology that I think makes the manuscript not proper for publishing:

1- The sample size doest seem to be adequate.

2- The data has been gathered between 2011 and 2022, that shows 10 years. Including covid-19 era. The subject of the study is in a way that time can effect on the results, and I think the data form this long period can not be analyzed together. So I think the manuscript is not proper for publishing.

3- The title can be improved.

4- Some sentences seem incomplete.

Manuscript needs minor english editing

Author Response

Dear Reviewer 3,

Thank you for your valuable comments regarding our manuscript entitled “The influence of gender equity in nursing education program on nurse job satisfaction” to be considered for publication in the special issue “Global Perspectives on Nursing and Midwifery Workforce Development” of Healthcare.

We greatly appreciate your comments. As you have suggested, we have revised our manuscript in the following manner.

  1. We appreciate your comment regarding the sample size. We have recalculated the minimum sample size using the G*power software and revised the manuscript with the correct information. We used effect size of 0.15, an alpha value of 0.05, a power of 0.80 and 12 predictors. The calculated minimum sample size was 127, and considering 20% drop-out rate, the estimated minimum sample size was 153.
  2. We apologize for the typo in the first submitted manuscript. The data collection period was from November 2021 to April 2022. We have revised the method section of the manuscript accordingly.
  3. Thank you for your suggestion regarding the title of the paper. We have revised the title to “The influence of gender equity in nursing education program on nurse job satisfaction.”
  4. We appreciate your comments regarding the overall English editing of the manuscript. We have consulted with a professional English editor, had our original manuscript reviewed and edited, and revised some of the incomplete sentences.

We believe that this manuscript is appropriate for publication in the special issue “Global Perspectives on Nursing and Midwifery Workforce Development” of Healthcare, as it provides scientific evidence of gender equity in nursing education program and how it associated with nursing workforce.

The authors of this manuscript comply with the ethics policies of the Healthcare as stated in the Healthcare Author Guidelines on its website. This manuscript has not been published elsewhere nor simultaneously submitted for publication elsewhere. We have no conflicts of interest to disclose. The manuscript has been read and approved for submission by all contributing authors.

Thank you for your time.

Round 2

Reviewer 3 Report

The revisions are acceptable.